# A Review of the Discriminant Analysis Methods for Food Quality Based on Near-Infrared Spectroscopy and Pattern Recognition

**DOI:** 10.3390/molecules26030749

**Published:** 2021-02-01

**Authors:** Jian Zeng, Yuan Guo, Yanqing Han, Zhanming Li, Zhixin Yang, Qinqin Chai, Wu Wang, Yuyu Zhang, Caili Fu

**Affiliations:** 1College of Electrical Engineering and Automation, Fuzhou University, Fuzhou 350108, China; zengjian_2868@163.com (J.Z.); orianaguo@gmail.com (Y.G.); wangwu@fzu.edu.cn (W.W.); 2Beijing Key Laboratory of Flavor Chemistry, Beijing Technology and Business University (BTBU), Beijing 100048, China; caili.fu@nusri.cn; 3Ministry of Education Key Laboratory of Medical Instrument and Pharmaceutical Technology, Fuzhou University, Fuzhou 350108, China; 4National University of Singapore (Suzhou) Research Institute, 377 Lin Quan Street, Suzhou Industrial Park, Suzhou 215123, China; yanqing.han@nusri.cn (Y.H.); lizhanming@just.edu.cn (Z.L.); yang_zhixin12138@163.com (Z.Y.); 5Key Laboratory of Eel Culture and Processing of Fujian Province, Fuzhou 350208, China

**Keywords:** near-infrared spectroscopy, food quality, pattern recognition, deep learning, qualitative analysis

## Abstract

Near-infrared spectroscopy (NIRS) combined with pattern recognition technique has become an important type of non-destructive discriminant method. This review first introduces the basic structure of the qualitative analysis process based on near-infrared spectroscopy. Then, the main pretreatment methods of NIRS data processing are investigated. Principles and recent developments of traditional pattern recognition methods based on NIRS are introduced, including some shallow learning machines and clustering analysis methods. Moreover, the newly developed deep learning methods and their applications of food quality analysis are surveyed, including convolutional neural network (CNN), one-dimensional CNN, and two-dimensional CNN. Finally, several applications of these pattern recognition techniques based on NIRS are compared. The deficiencies of the existing pattern recognition methods and future research directions are also reviewed.

## 1. Introduction

With frequent exposure of the adulteration of food in the market, food safety issues have raised global health concerns. Adulterated food not only harms human health but also severely disrupts the order of the food market. Therefore, the detection and discrimination of food quality is of great importance. Traditional food quality detection methods are mainly based on high-performance liquid chromatography [1], mass spectrometry [2], or DNA technique [3]. However, the applications of these methods are limited by the following disadvantages: they are time-consuming, have low efficiency, have high measurement cost, require massive expertise, and cause damage to samples. Thus, it is necessary to seek a fast and efficient detection tool to facilitate the detection process and widen the applications. Near-infrared spectroscopy (NIRS) technology was developed nearly half a century ago as a detection tool. It obtains feature information of the hydrogen-containing groups (O-H, N-H, C-H) in organic substances by scanning the near-infrared spectrum of the samples [4] and possesses many advantages including being faster, having more accuracy, and being non-destructive. Recently, near infrared hyperspectral imaging (NIRHSI) technology was developed to obtain the chemical spectral information and the spatial distribution of chemical substances in heterogeneous samples simultaneously [5]. It should be noted that by extracting the region of interest (ROI) from the hyperspectral image of the sample and calculating the average spectrum of each band within the ROI, the NIRHSI data can be reduced to near-infrared spectral data [6]. Thus, in this paper, we focus on the discrimination methods based on near-infrared spectroscopy (NIRS). NIRS is an indirect analysis technique that requires the establishment of empirical models through suitable pattern recognition methods. Although discrimination methods based on NIRS have been widely used for food quality discrimination, there are still some issues to be solved. In this review, the current developed NIRS-based food discrimination methods are compiled. The organizational structure of this article is shown in Figure 1, including pretreatment methods, traditional pattern recognition methods, and deep learning methods. The basic procedures of qualitative analysis based on NIRS is introduced in Section 2. Section 3 describes the main pretreatment methods used in NIRS. Section 4 presents different traditional pattern recognition methods combined with NIRS and their applications in food discrimination. Section 5 shows the potential of combining deep learning methods with NIRS to applied to food quality analysis. Finally, Section 6 summarizes the whole review and some future research directions are discussed.

## 2. Basic Structure of Qualitative Analysis Process Based on NIRS

The qualitative analysis process includes NIRS measurement and chemometric method for spectral analysis, which is shown in Figure 2.

Firstly, in order to obtain NIRS data, the samples are collected, stored, and specially processed if necessary. Then, each sample is measured with a near-infrared spectrum instrument. The obtained spectrum range is usually between 4000 and 10,000 cm^−1^. Although the near-infrared spectrometer is equipped with a high-performance detection device, the collected data can still be affected by the noise of the measuring instrument, thus affecting the effect of modeling [7]. Therefore, the spectral data must be pretreated before using chemometric methods. Secondly, the spectral peaks of different samples are usually similar in shape and position. The peak overlaps and interaction in some segments can be severe, which makes it difficult to distinguish the samples directly. Therefore, it is necessary to use chemometrics methods to achieve spectrum analysis and discrimination. Thirdly, the spectrum obtained has thousands of dimensions, which brings challenges to chemometrics methods, such as the problem of “the curse of dimensionality”. Therefore, feature extraction methods usually are adopted to reduce the dimensions of the spectrum, and the discrimination is achieved by the establishment of pattern recognition methods.

Clearly, preprocessing methods, feature extraction methods, and pattern recognition methods are three key procedures of qualitative analysis. Thus, preprocessing methods, feature extraction methods, and pattern recognition methods for food discrimination are reviewed in the following sections.

## 3. Pretreatment Methods

When analyzing complex samples, the final qualitative analysis results are often affected by interference factors such as stray light, noise, and baseline drift [8]. Therefore, in order to increase the stability and robustness of the qualitative analysis model, it is necessary to preprocess the original spectrum to eliminate the influence of uncertain factors and useless information variables before modeling [9]. There are many kinds of spectral preprocessing methods. According to their effect on preprocessing, they can be divided into four categories: baseline correction, scatter correction, smoothing treatment, and characteristic scaling [10,11]. Baseline correction is used to eliminate the influence of instrument background or drift. This method includes first derivative (D1), second derivative (D2), and continuous wavelet transformation. Scattering correction methods, such as multiplicative scatter correction (MSC) and standard normal variables (SNV), are used to avoid the influence of scattering caused by uneven particle distribution or different particle sizes to enhance the absorbance information that most related to the spectral content in the spectrum. Savitzky-Golay smoothing is a low-pass filter. It can effectively filter out the high-order signal interference and remove the effects of random noise. Feature scaling can eliminate the adverse effects caused by large-scale differences. This method includes mean centering, min-max scaling, and standardization.

## 4. Traditional Pattern Recognition Method

Traditional pattern recognition methods can be divided into supervised learning and unsupervised learning. The main difference between these two categories is that supervised learning requires the tag information of the samples when training the models, whereas this is not necessary for unsupervised learning. Supervised learning mainly applies some shallow learning machines to achieve classification, while unsupervised learning mainly focuses on the clustering analysis. The following subsections describe them in detail.

### 4.1. Shallow Learning Machines

Shallow learning machines lack effective feature expression ability. This means that the performance of these models highly depends on valid feature extraction. By combining feature extraction methods with shallow learning machines, the problem of curse of dimensionality can be avoided and the performance of shallow learning machines can be greatly improved [12]. The commonly used feature extraction methods include principal component analysis (PCA) [13,14,15,16], successive projections algorithm (SPA) [17,18], uninformative variables elimination (UAE) [19], and intelligent optimization algorithm [4,20,21]. The commonly used shallow learning machines include partial least squares-based discriminant analysis (PLS-DA) [22], soft independent modeling of class analogy (SIMCA), support vector machines (SVM), back propagation artificial neuron network (BP-ANN), and other discriminant analysis methods. The following parts will introduce some shallow learning machines according to their types.

#### 4.1.1. PLS-DA

PLS-DA is a supervised qualitative analysis method based on PLS regression analysis [23,24]. In order to achieve classification, the output matrix of PLS-DA is a one-hot encoding, while the input matrix and output matrix of PLS regression are continuous variables. For example, the PLS-DA output matrix Y = [y_1_, y_2_, y_3_] of a three-category task is shown in Table 1. The threshold θ_i_ is set to determine the category. If y_i_ = max(y_1_, …, y_n_) > θ_i_, the predicted label is the i-th category.

The PLS-DA model is widely applied in qualitative analysis of food NIRS, such as the identification of geographical origins and varieties. In the field of food geographical origin identification, in [15,25], PCA and PLS-DA binary classification models were built to discriminate the geographical origins of Huajiao and French olive oils. The accuracy rates of the model were 97.6% and 100%, respectively. In Reference [26], a PLS-DA classification model was established to trace the origin of Chilean flour and wheat grain. The flour possessed the best matrix for PLS-DA and the accuracy rate exceeded 90%. In Reference [27], PCA method combined with different preprocessing methods and different spectral bands was used to establish PLS-DA binary classification models to identify the origins of rice. The comparative experimental results showed that the model combining SNV and 4000–5500 cm^−1^ spectral band was the most suitable, and the accuracy of the training set and testing set were 93.33% and 86.67%, respectively.

In the field of food varieties identification, in Reference [28], multiple PLS-DA models were established by combining different preprocessing methods to achieve the authentication of Iberian pork. The experiment showed that the model in combination with the SNV and D2 initialization methods achieved 93.2% and 93.4% accuracy in the training set and testing set, respectively, of the fresh meat and subcutaneous fat samples. In Reference [29], on the basis of visible NIRS, the authors constructed a PCA and PLS-DA model to identify varieties of Riesling and Chardonnay commercial wine. The recognition rate of Riesling and Chardonnay wines was up to 100% and 96%, respectively. Similarly, PLS-DA model was established to identify oat and grain kernels through combining with NIRHSI technology. The results showed that the correct classification rates of oat and grain kernels were 97.1% and 99.0%, respectively, proving the effectivity of this method [30]. In Reference [31], on the basis of NIRHSI technology, the authors established SVM and PLS-DA models with a reduced set of only 54 wavelengths, resulting in excellent accuracy rates with 98.29% and 99.00% for the prediction sets, respectively. In Reference [32], on the basis of the spectral range of 5000–4000 cm^−1^, SNV method, and eight principal components, the PLS-DA model showed the best performance in the assessment of hard seed characteristics of legume seeds, with an accuracy of 100%. Furthermore, PLS-DA models were also applied to meat quality identification [33], obtaining good results.

#### 4.1.2. SIMCA

SIMCA is a supervised qualitative analysis method based on PCA algorithm [34,35]. The establishment of SIMCA model consists of three steps. First, PCA algorithm is used on the calibration set and the principal component regression model is established. Second, the similarities among the samples of different classes are calculated to obtain the critical value θ_0_ Finally, the similarity θ_i_ among the tested sample and the samples of the given category in the calibration set is calculated. If θ_i_ ≤ θ_0_, the tested sample belongs to this category; otherwise, it does not belong to this category. In general, the classification performance of the SIMCA model is evaluated by
(1)$${\mathrm{Recognition}\;\mathrm{rate}}_{i}=\frac{n_{c}}{N_{i}}$$
(2)$${\mathrm{Rejection}\;\mathrm{rate}}_{i}=\frac{n_{r}}{N-N_{i}}$$
where N_i_ is the number of samples in the i category, n_i_ is the number of samples that correctly identify the category, n_r_ is the number of samples rejected from other categories, and *N* is the total number of samples.

In the field of food geographical origin identification, in References [36,37], SIMCA models with different numbers of principal components, different pretreatment methods, and different near-infrared spectral bands were established to identify wolfberry (*Lycium barbarum* L.) from eight different origins, achieving good results. In References [38,39], the SIMCA models were applied to identify the geographic origins of *Forsythia suspense* and mutton. The recognition rate of the geographic origin of forsythia was 90%, while the average recognition rate and rejection rate of the geographic origin of lamb were 93.75% and 100%, respectively.

In the field of qualitative analysis of food based on NIRS, the SIMCA model is also very popular. In Reference [40], full-band near-infrared spectroscopy data combined with SIMCA technology were used to identify three brands of milk powder—the recognition rate was 84.3% and the rejection rate was 91.6%. In Reference [41], combined with the SNV and D2 pretreatment methods, the SIMCA and PLS-DA models were established to identify the brand traceability of edible vinegar. The results showed that SIMCA was more competitive than PLS-DA, with accuracy rates of 84% and 94.7%, respectively. In Reference [42], the full spectrum band and the optimized characteristic band were used to establish the SIMCA models to classify four different types of wheat. The results showed that the recognition rate and rejection rate of the model were both 100%. In addition, the PLS-DA and SIMCA were also successfully applied to qualitative analysis of different navel orange varieties [43], with both recognition rates reaching 100%.

#### 4.1.3. SVM

SVM was firstly proposed by Vapnik in 1995 and is based on statistical learning theory to solve two-group classification problems [44]. It is then extended to solve multiple classification problems in Reference [45]. By using the principle of structural risk minimization and Vapnik-Chervonenkis (VC) dimension theory, SVM can handle the data of high dimensions with good generalization and accuracy [46]. The VC dimension for the set of functions {F} is defined as the maximum number of training points that can be shattered by {F} [47]. It should be noted that SVM can only basically reflect the linear relations between the inputs and outputs. For nonlinear applications, the kernel function should be firstly used to map the data to high dimension, and then the optimal hyperplane should be constructed to classify the linearly separable data.

Since the mechanism of molecular vibration under different frequencies of light irradiation is complex, the obtained near-infrared spectral data contains nonlinear relations between the spectrum and its category. Thus, the SVM mentioned here is kernel SVM. The commonly used SVM kernel functions include linear kernel, polynomial kernel, radial basis function (RBF), and Gaussian kernel. Among these SVM kernels, RBF kernel is most widely used due to its few hyperparameters and fast approximation speed [48]. For SVM, the penalty parameter c and the RBF kernel function parameter g are two key parameters that influence the performance of the SVM model. The parameter c represents the tolerance to errors, which determines the generalization ability of the SVM model. The value of g influences the training and prediction speed of the SVM model [49]. However, in real applications, appropriate combination of these two parameters is difficult. Therefore, the way in which to optimize the two parameters is still a considerable issue. The commonly used methods for the parameter optimization of SVM model include trial and error method [50,51,52], grid search algorithms, and intelligence optimization algorithms.

The trial-and-error method-based SVM has many successful applications. For example, the RBF kernel function-based SVM models were established for qualitative analysis of apple origin and species [50], soybean oil color discrimination [51], and authenticity identification of tea [52], with accuracy rates of 100%, 100%, and 84.44%, respectively. However, the trial-and-error method relies on large numbers of experiments and expert experience. Moreover, the result searching may fall into local optimization, which will limit the performance of the model.

To avoid the drawbacks of the trial-and-error method, researchers have used the grid search method to select the c and g parameters. Established SVM models had been successfully applied to identify resin [53], adulterated peanut oil [54] origins, and varieties of maize kernels [14]. The model accuracy rates were 100%, 100%, and 95%, respectively. Although the grid search method is simple, universal, and easy to implement, the grid search algorithm is an exhaustive method. Thus, when the search range is relatively large, there will be too many invalid operations, which will result in a slow search speed.

Different from the grid search method, the intelligent optimization algorithm can select optimal parameter combination through automatic searching. The intelligent optimization algorithm has higher search efficacy and robustness as it is not easily affected by the initial value. It can also generally avoid falling into the local optimization. As a result, it has obtained more and more attention in the SVM training process in recent years. For example, as reported in Reference [55], genetic algorithm (GA) was used to select c and g parameters and then the SVM model combined with the SNV preprocessing method was established to identify different vegetable oils. Furthermore, the particle swarm optimization (PSO) was used to determine the c and g parameters for the establishment of the PCA-SVM model in order to identify the adulteration of hops [56] and Huajiao powder [57]. Apart from the abovementioned algorithms, the newly developed intelligent optimization algorithms, such as differential evolution algorithm, ant colony optimization algorithm, whale optimization algorithm, and wolf optimization algorithm can also be used for parameter optimization. In real applications, the algorithm complexity and the search efficiency should be taken into consideration. 

Furthermore, there have been some improved SVM models used in the qualitative analysis of foods performed through NIRS. Least squares SVM (LS-SVM) is the most typical one. It converts inequality constraints into equality constraints on the basis of SVM. It can also transform the quadratic programming problem into the linear equation solution in SVM. Thus, compared with traditional SVM algorithm, the LS-SVM algorithm is simpler and faster to realize and is more accurate [58]. In Reference [59], an LS-SVM model with GA optimization was established to identify whether the chestnut was moldy or not, with an accuracy of 93.56%. In Reference [60], an error-correcting output coding SVM model using NIRHSI technology for of six different commercial tea products is presented, and an excellent accuracy rate of 97.41% was obtained. In References [61,62,63], PLS-DA, SIMCA, BP-ANN, SVM, and LS-SVM models were established separately to identify honey adulteration, lotus seed powder, and its adulteration and damage of Huping jujube. Comparison studies showed that the performance of the LS-SVM model was more competitive than other traditional pattern recognition methods in the application of NIRS in food qualitative analysis. Moreover, milk adulteration identification models were built by two improved and simplified *k*-nearest neighbors of non-linear supervised pattern recognition methods and improved SVM algorithms in Reference [64]. Experiments showed that the calculation burden of the improved SVM model was reduced by 90%, and good results were obtained.

#### 4.1.4. Discriminant Analysis Method

According to different discriminant principles, discriminant analysis methods include Fischer discriminant analysis, distance discriminant analysis, and Bayes discriminant analysis. Discriminant analysis methods are widely used in qualitative analysis of NIRS data of foods.

##### Fisher Discriminant Analysis

Fischer discriminant analysis, also known as linear discriminant analysis (LDA) method, is a projection-based method. By choosing appropriate projection axes for LDA, the projection points of similar samples are close to each other, while the projection points of different classes are as far as possible [65,66]. Then, the category of each sample is determined according to its distance from the projection points of other samples. 

In the field of qualitative analysis of NIRS data, in Reference [13], PCA-LDA, SVM, and SIMCA methods were established to identify Chinese liquor, including 22 types, 10 famous brands, and 6 flavor types. Experiments showed that PCA-LDA was the best model. The average accuracy of the training set was 98.94%, and the average accuracy of the test set was 95.70%. In Reference [67], combined with the artificially selected near-infrared spectral bands, LDA and SIMCA were used to establish various discrimination methods of extra virgin olive oils. The experimental results showed that the average accuracy rates of the fivefold cross-validation were 90.5% and 80.5%, respectively. In this case, it was clear that LDA had better performance. In another case [68], PCA-LDA and GA-LDA were used to establish the identification model of macadamia cultivars. Experiments showed that, with Savitzky–Golay smoothing preprocessed spectra, the accuracy of GA-LDA was higher than 94.44%.

##### Distance Discriminant Analysis

Since the labels of the training data are known, the distance between the center of the unknown data and the training data can be calculated to determine which category the sample belongs to. Mahalanobis distance is usually used in distance discriminant analysis. The Mahalanobis distance discriminant analysis (MDDA) is used to find the Mahalanobis distance of the test sample from the center of each category. The category with the smallest distance is selected as the category identified by the MDDA model [69]. 

In the field of food quality evaluation and adulteration discrimination, the authors of [70] used the MDDA model to evaluate rice quality, and the accuracy of the model was 98.33%. Moreover, in References [71,72], MDDA combined with PCA was used to build qualitative models for peanut seed quality determination and surimi grade estimation—both of the models obtained good identification results. MDDA, SIMCA, and LS-SVM algorithms were compared in References [13,73] for estimating plum maturity levels and quality of Chinese liquors. The experimental results demonstrated that the MDDA and PCA-MDDA models had the best performance—the accuracy rates of testing set were 95.7% and 96.3%, respectively. In addition, in the tracing of food origin, the MDDA method is also involved. For example, the identification of sea cucumber geographical origin [74] and the discrimination of Chinese wolfberry from Zhongning [75]. The accuracy rates reached 100% in both cases.

##### Bayesian Discriminant Analysis

Bayesian estimation is to obtain the probability density function by combining the new evidence with the prior probability, so as to maximize the posterior probability. In Bayesian discriminant analysis methods, the probability density functions of each of the categories are estimated first, and then certain classification rules are set to determine the category for each test samples. Usually, the sample belongs to the category that has the greatest probability. Typical Bayesian discriminant analysis methods include the relevance vector machine (RVM) and the naive Bayesian classifier (NBC).

In the field of qualitative analysis of NIRS data, RVM and new clustering algorithm were used in Reference [76] to establish various identification models of adulteration of *Ganoderma lucidum* spore oil. The accuracy of the model was 100%. In Reference [77], on the basis of the competitive adaptive reweighted sampling (CARS) wavelength optimization, the authors compared RVM, SVM, and LS-SVM identification models of traditional Chinese medicine Sanyeqing and its similar species. The experimental results indicated that CARS-RVM achieved the best performance. The accuracy of the correction set and the verification set were all 100%. In addition, in Reference [78], an improved naive Bayesian classifier (INBC) combined with a fast search and find of density peaks (CFSFDP) clustering method was proposed to identify the origin of the traditional Chinese medicinal Sanyeqing. It should be pointed out that INBC-CFSFDP method can not only identify the known origin in the database accurately, but also identify the unknown origin. The recognition rate of unknown and known geographical origin in the database was 100%. This research opened up a new horizon for qualitative analysis of NIRS.

#### 4.1.5. Artificial Neural Network

Artificial neural network (ANN) is a mathematical model based on the mechanism of the processing of complex information by a biological neural network. ANN generally includes three layers: the input layer, hidden layer, and output layer. In the qualitative analysis of NIRS, the features extracted from the spectral data are inputted into the input layer. The number of input units is consistent with the input dimension of the spectral data. The hidden layer is to construct the relationships between the input features and the output categories. The outputs of the hidden layer are real numbers. The output layer is equipped with discriminant principles to generate the classified categories. The values of the output layer are the category labels. Generally, the more hidden layer units, the stronger nonlinear ability. However, too many hidden layers will increase the risk of overfitting. In the qualitative analysis of NIRS, the BP-ANN, RBF-ANN, and extreme learning machine (ELM) models are most widely used. 

In terms of qualitative analysis of NIRS data, PCA combined with BP-ANN methods were applied in References [79,80] to establish the identification models of cultivated and wild *Erigeron breviscapus* and apple varieties. The recognition rate reached 100% in both cases. In Reference [81], in order to establish the BP-ANN model, NIRHSI technology was used to obtain the NIRS data. The model was used to identify soybean varieties. The training accuracy rate reached 97.50% and the average test accuracy rate was over 93.88%. The MDDA and BP-ANN models were compared in Reference [82] on the basis of NIR data of the browning of plum fruit. Experiments showed that both models can effectively identify the browning of plum fruit, and the recognition rates were all 100%. The PCA-BP-ANN and PLS-BP-ANN models were compared in Reference [83] on the basis of NIR data of different species of fish, and the accuracy of training set and testing set were 96.4% and 95.5%, respectively. The PLS-DA and BP-ANN models were compared in Reference [84] on the basis of NIR data of camellia oil from different geographical origins. Experiments showed that the accuracies of MDDA and BP-ANN models were similar, but both were better than the PLS-DA model. The PCA-BP-ANN model had the best performance, with an accuracy rate of 95.83%. Combining the smoothing pretreatment method with the models can help to improve the models’ performances. It is worth noting that the performance of ANN greatly depends on its parameters. Thus, how to optimize the methods’ parameters is widely studied. For example, in Reference [85], BP-ANN was optimized by PSO method. The spectral band and parameters were optimized to establish the authenticity discrimination model of official rhubarb samples. The model accuracy reached 96.15%. Different from ANNs, the number of neurons in the hidden layer is the only parameter needed to be determined in ELM. Thus, ELM contains less parameters to be optimized, which means the model training process is simplified. The ELM combining with the CFSFDP clustering method was used to identify the adulterated grape seed oils in Reference [86], and a detection system was designed, with an accuracy rate of 100%. 

#### 4.1.6. Other Qualitative Analysis Methods

For qualitative analysis of food by NIRS, in addition to the above-mentioned traditional pattern recognition methods, there are some other methods that are worth noting. For example, random forest (RF) and PCA were used to establish the identification model of green tea species on the basis of the joint information from NIRS and ultraviolet-visible spectroscopy [87]. The results show that the RF model can effectively identify five green tea varieties, and the accuracy rate of the model was 96%. Similarly, in Reference [88], Fourier transform mid-infrared spectroscopy and NIRS technologies were combined with single spectra analysis and multi-sensor information fusion strategy to establish the RF model for the origin identification of Panax notoginseng. The accuracy of RF identification model was 95.6%, suggesting an effective identification ability to identify Panax notoginseng from five different origins. In Reference [89], *k*-nearest neighbors (KNN) and LDA were used to establish the identification model of purple sweet potatoes, white sweet potatoes, and their adulterate samples. The prediction accuracy of LDA and KNN were 100% and 97.4%, respectively. In addition, HCA, KNN, and LDA were used to identify four production sources of tomato, and good results were obtained [90]. In Reference [91], NIRS data combined with PCA were used to establish SVM and learning vector quantization (LVQ) neural networks for coix seed variety identification, and the prediction accuracies were 100% and 90.91%, respectively. In Reference [92], PLS-DA, KNN, PCA, and classification and regression trees (CART) models were established to identify the quality of frying oil. The experimental results showed that the proposed models could effectively identify qualified and unqualified frying oil.

### 4.2. Clustering Analysis

The cluster analysis method is an unsupervised pattern recognition method without labeled matrix. It can automatically realize classification by calculating the similarity of the samples. For NIRS data processing, the distance of the spectrum [93] is generally used to calculate the similarity of the data. However, for the processing of NIRS data of high-dimensions, distances among samples from different catalogs could be very close or the same. Usually, the feature extraction methods introduced in Section 4.1 are firstly used to eliminate overlapping parts of information and reduce the dimensions of NIRS. Then, the cluster analysis methods are applied. 

The cluster analysis methods are widely used in qualitative analysis of NIRS data for food because they save the work of labeling data. For example, in Reference [94], with the 1100–1800nm band and the D2 and SNV pretreatment methods, the authors used the PCA method to establish the qualitative analysis model of shelf-life of Mopan persimmon. The accuracy of the model training set and testing set were 95.4% and 93.3%, respectively. In the PCA score plots, the visual classification can be presented directly, and samples with similar spectral characteristics can be clustered. In References [95,96,97], hierarchical cluster analysis (HCA) with different wavelength bands and different pretreatment methods were utilized to analyze chicken origins, identify sesame oil and its adulterations, and discriminate the adulterated Cheng’an strawberry. Compared with PCA, HCA had better accuracy, with the accuracy rates of the HCA model being 100%, 100%, and 93.3%, respectively. In addition, other clustering methods, such as a fast search and find of density peaks clustering method, were also involved [78,86].

## 5. Deep Learning Methods

Deep learning has become a hot research direction in the field of artificial intelligence and has developed rapidly. Convolutional neural network (CNN) is one of the representative algorithms of deep learning. Like ANN, the purpose of CNN is also to simulate the learning process of the human brain. However, the structure of each layer in CNN is different from that of ANN. For example, the hidden layer of CNN involves several convolutional layers and pooling layers to automatically extract the essential features from the data. The fully connection layer realizes fitting or classification. In a word, the deep-learning-based model is an end-to-end self-adaptive model that only needs the input data and output data, and then the classified results can be obtained. The internal model can complete the feature extraction and classification process simultaneously.

Due to the excellent learning abilities for high-dimensional data, CNN has achieved great success in the food qualitative analysis with NIRS data. For example, a six-layer CNN model was designed in Reference [98] to achieve the quality identification of three types of macadamia nuts, with an accuracy of 100%. It provided a reliable technical support for the efficient and low-cost quality determination of macadamia nuts. In Reference [99], a 1D convolutional neural network (1D-CNN) was proposed for the discrimination of aristolochic acids and their analogues in traditional Chinese medicine. The validation set accuracy of 1D-CNN, PCA-SVM, T-distributed stochastic neighbor embedding support vector machine (*t*-SNE-SVM), SVM and BP-ANN were 100%, 91.41%, 90.63%, 87.89%, and 39.45%, respectively. The comparison results showed that traditional shallow learning machines heavily relied on the feature extraction algorithms such as PCA or *t*-SNE, while the accuracy of the 1D-CNN model was still very competitive without using any feature extraction algorithms, and was much better than BP-ANN. Similarly, SVM and self-adaptive evolutionary ELM and CNN models were established to identify the different geographical origins of *Tetrastigma hemsleyanum*. The accuracy rates were 90%, 98.3%, and 100%, respectively [100]. Experiments showed that CNN was faster and more accurate than traditional methods. CNN was also successfully used in the field of tobacco geographical origin identification. In Reference [101], the improved LeNet-5 network structure was used to identify the three major tobacco leaf production areas. The accuracy of the training set and testing set were 98.6% and 95.0%, respectively. In Reference [102], 1D-CNN, 2D-CNN, and PLS-DA were established to identify the geographical origin of tobacco leaves. Experiments showed that the performance of 1D-CNN and 2D-CNN was better than that of PLS-DA. The model accuracy rates were found to be 93.15%, 93.05%, and 81.25%, respectively. Although at present there are only a few studies on the application of CNN in the field of qualitative analysis of NIRS data of food, it is worthy of further research. Examples of all the mentioned applications of pattern recognition technology for quantitative analysis of food based on NIRS in this review are shown in Table 2.

## 6. Conclusions

Pattern recognition methods based on NIRS technique are widely used in the detection of food quality. The qualitative detection methods mainly focus on the early period shallow learning machines. However, they are not suitable for the feature learning of complex inputs when the number of samples is much smaller than the number of feature dimensions. Therefore, traditional NIRS technique-based discrimination methods usually employ feature extraction methods to eliminate redundant wavelengths before constructing shallow learning classifier. Still, the feature extraction methods depend on practical experience and professional knowledge. In addition, the selected features belonging to a certain kind of dataset may be not suitable for another dataset. This means that the features need to be re-extracted when the tested food products are replaced, which consumes a large amount of time. To avoid these drawbacks, deep learning theories such as CNN models are developed to achieve qualitative analysis. It can automatically extract important features from large collections of high-dimensional raw data with high recognition performance. Therefore, the CNN models based on NIRS techniques have great potential in the field of qualitative analysis of food. 

The deep learning methods have excellent performance in the analysis of large datasets. If the dataset is small, it may lead to over fitting problem of the model. However, in the application of near infrared spectroscopy, a large number of samples are often difficult to obtain. For example, due to the long growth cycle and limited environmental conditions of some samples, the cost of obtaining a large number of NIRS data is too high. Therefore, the way in which to apply deep learning to the analysis of NIRS datasets with sample size far less than the sample dimension is a challenge in the future.

In addition, pattern recognition methods have developed from traditional shallow learning machines to deep learning methods. The novel methods have obtained effective results in food quality detection. However, the theories are limited to known categories that have been trained in the training stage. New unknown categories that have never merged in the dataset are less considered. Clearly, it means that the training set-based models cannot be applied to new unknown categories before training. In reality, as driven by interests, products of the same species but cultivated by new cultivation technology or new cultivation environment will continue to emerge, yet the qualities of these products may be unknown. Thus, enhancing the unknown categories recognition ability of the qualitative analysis models is a key direction for future research.

## Figures and Tables

**Figure 1 molecules-26-00749-f001:**
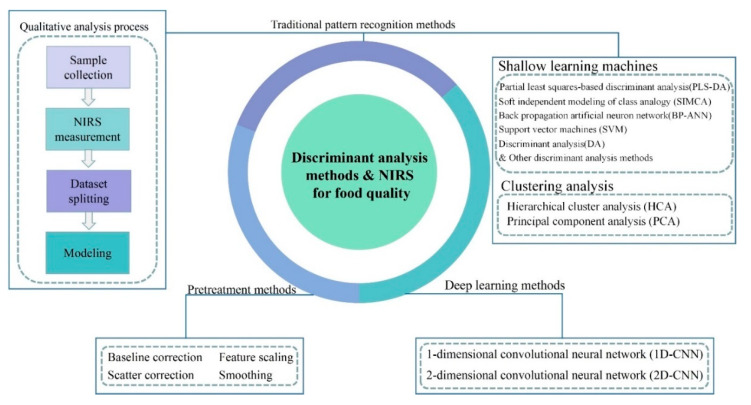
The organizational structure of the article.

**Figure 2 molecules-26-00749-f002:**
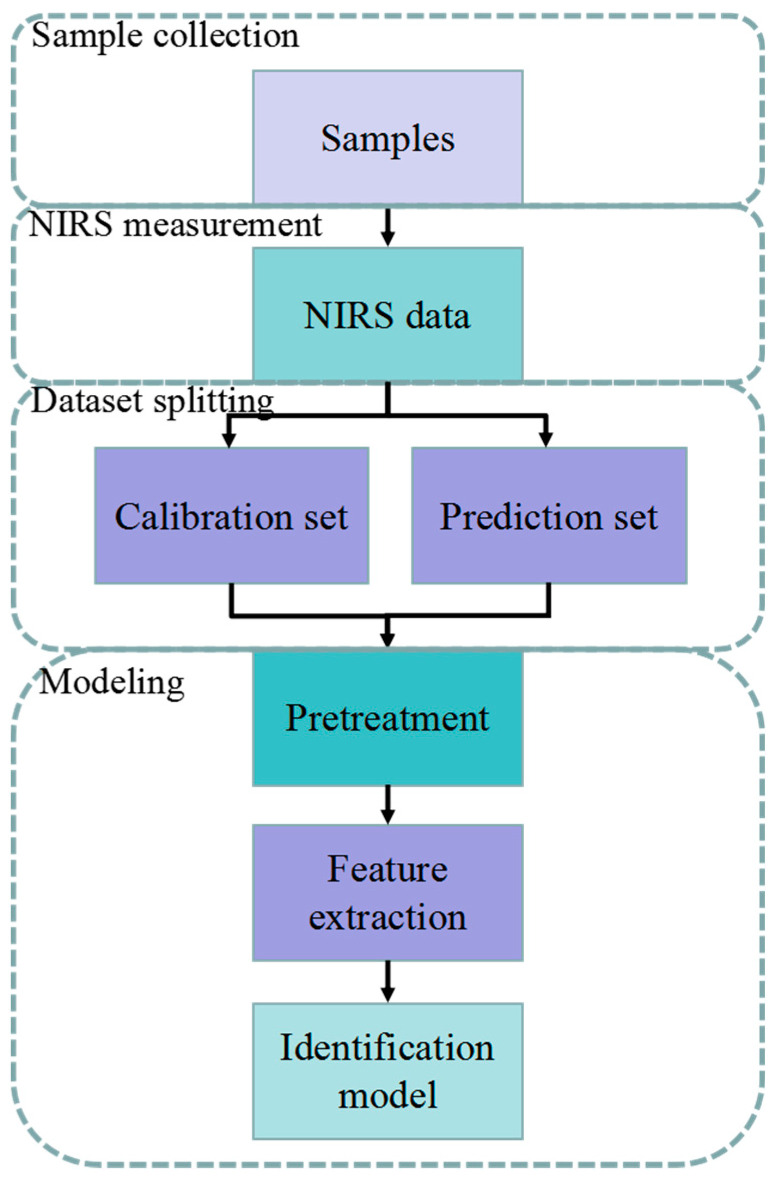
Process of the qualitative analysis by near infrared spectroscopy (NIRS).

**Table 1 molecules-26-00749-t001:** The partial least squares-based discriminant analysis (PLS-DA) output matrix of a three-category task.

Category	Y		
A	1	0	0
B	0	1	0
C	0	0	1

**Table 2 molecules-26-00749-t002:** Application of pattern recognition technology for the qualitative analysis of food.

Pattern Recognition Technology		Applications	Objectives	Source
Shallow learning machine	PLS-DA	Geographic origin identification	Huajiao; French olive oils; rice; Chilean flour and wheat grain	[15,25,26,27]
		Quality identification	Iberian pork; meat	[28,32]
		Variety identification	Commercial wine; oat/grain kernels; hard seeds of legume plants	[29,30,33]
	SIMCA	Geographic origin identification	Wolfberry; *Forsythia suspense*; mutton	[36,37,38,39]
		Brand traceability	Milk powder; edible vinegar	[40,43]
		Defective items classification	Wheat grain	[42]
		Variety identification	Navel orange	[43]
	SVM	Origin and variety analysis	Apple; maize kernels; hops	[14,50,56]
		Color discrimination	Soybean oil	[51]
		Authenticity/adulteration identification	Tea; peanut oil; Huajiao powder; honey; lotus seed powder; milk	[52,54,57,61,62,64]
		Defective items classification	Chestnut; Huping jujube	[59,63]
		Classification	Tea	[60]
	Discriminant analysis	Identification of brand and type	Chinese liquor	[13]
		Variety identification	Extra virgin olive oils; Sanyeqing	[67,77]
		Nondestructive classification	Macadamia cultivars	[68]
		Geographic origin identification	Sea cucumber; Chinese wolfberry from Zhongning; Sanyeqing	[74,75,78]
		Quality and grade estimation	Peanut seed; surimi; plum	[71,72,73]
		Adulteration identification	*Ganoderma lucidum* spore oil	[76]
	Artificial neural network	Variety identification	*Erigeron breviscapus*; soybean; apple; fish	[79,80,81,83]
		Color discrimination	Plum	[82]
		Geographical origin	Camellia oil	[84]
		Authenticity/adulteration discrimination	Rhubarb; grape seed oils	[85,86]
	Others	Variety identification	Sweet potatoes; green tea; coix seed	[87,89,91]
		Origin identification	Panax notoginseng; tomato	[88,90]
		Quality assessment	Frying oil	[92]
Clustering analysis		Shelf-life analysis	Mopan persimmon	[94]
		Geographic origin analysis	Chicken	[95]
		Adulteration identification	Sesame oil; Cheng’an strawberry	[96,97]
Deep learning methods		Quality identification	Macadamia nuts	[98]
		Discrimination	Aristolochic acids and their analogues in traditional Chinese medicine	[99]
		Geographical origin	Sanyeqing; tobacco leaves	[100,101,102]

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
