# Peer review of "A Review of the Discriminant Analysis Methods for Food Quality Based on Near-Infrared Spectroscopy and Pattern Recognition"

_molecules, 2021, doi:10.3390/molecules26030749_

Round 1

Reviewer 1 Report

The authors provide a nice and interesting review. In this paper, the currently  discriminant analysis methods for food quality based on near-infrared spectroscopy and pattern recognition are reviewed.

The review discusses most of the key publications relevant to the topic. However I suggest including other well-known classification methods on NIRS data: K-Nearest Neighbors (KNN) method, Classification And Regression Tree (CART), Learning Vector Quantization (LVQ) and Discrimination of using NIR hyperspectral imaging.

I recommend including new challenges about the future of the research  in the Conclusion section.

Author Response

Response to Reviewer 1 Comments

Point 1: The review discusses most of the key publications relevant to the topic. However, I suggest including other well-known classification methods on NIRS data: K-Nearest Neighbors (KNN) method, Classification And Regression Tree (CART), Learning Vector Quantization (LVQ) and Discrimination of using NIR hyperspectral imaging.

Response 1: We sincerely appreciate this valuable suggestion and we have included K-Nearest Neighbors (KNN) method, Classification And Regression Tree (CART), Learning Vector Quantization (LVQ) and Discrimination into our revised manuscript in line 376-396. The text reads:

"4.1.6 Other qualitative analysis methods

For qualitative analysis of food by NIRS, in addition to the above-mentioned traditional pattern recognition methods, there are some other methods that worth noting. For example, the random forest (RF) and PCA were used to establish the identification model of green tea species based on the joint information from NIRS and ultraviolet-visible spectroscopy [86]. The results show that the RF model can effectively identify five green tea varieties, and the accuracy rate of the model was 96%. Similarly, in [87], Fourier transform mid-infrared spectroscopy and NIRS technologies were combined with single spectra analysis and multi-sensor information fusion strategy to establish the RF model for the origin identification of Panax notoginseng. The accuracy of RF identification model was 95.6%, suggesting an effective identification ability to identify Panax notoginseng from five different origins. In [88], K-nearest neighbors (KNN) and LDA were used to establish the identification model of purple sweet potatoes, white sweet potatoes and their adulterate samples. The prediction accuracy of LDA and KNN were 100% and 97.4%, respectively. In addition, HCA, KNN and LDA were used to identify four production sources of tomato and good results were obtained [89]. In [90], NIRS data combined with PCA were used to establish SVM and learning vector quantization (LVQ) neural networks for coix seed variety identification, and the prediction accuracy were 100% and 90.91%, respectively. In 91], PLS-DA, KNN, PCA and classification and regression trees (CART) models were established to identify the quality of frying oil. The experimental results showed that the proposed models could effectively identify qualified and unqualified frying oil."

Point 2: I recommend including new challenges about the future of the research in the Conclusion section.

Response 2: We appreciate this valuable suggestion and we have included new point in conclusion section. The text is in line 473-479 and read as:

"The deep learning methods have excellent performance in the analysis of large data sets. If the data set is small, it may lead to over fitting problem of the model. However, in the application of near infrared spectroscopy, a large number of samples are often difficult to obtain. For example, due to the long growth cycle and limited en-vironmental conditions of some samples, the cost of obtaining a large number of NIRS data is too high. Therefore, how to apply deep learning to the analysis of NIRS data sets with sample size far less than the sample dimension is a challenge in the future."

Reviewer 2 Report

Lines 68-72 – “Firstly, in order to obtain NIRSdata, the collected samples need to be specially processed to meet the  requirements  of near  infrared  spectrometer. Then  each  sample  is measured under specific temperature conditions to obtain spectral data. The obtained spectrum range is usually between 4000 and 10000 cm-1and has thousands of dimensions.” – the vast majority of NIR analyzes are non-destructive and do not require sample preparation. Also, specific temperature conditions is not a requirement for NIR analysis. Please change this sentence.

Lines 97-98 – “multiple scattering correction (MSC)” – MSC is multiplicative scatter correction.

Equations 1 and 2 are duplicated

Author Response

Response to Reviewer 2 Comments

Point 1: Lines 68-72 – “Firstly, in order to obtain NIRS data, the collected samples need to be specially processed to meet the requirements of near infrared spectrometer. Then each sample is measured under specific temperature conditions to obtain spectral data. The obtained spectrum range is usually between 4000 and 10000 cm-1and has thousands of dimensions.” – the vast majority of NIR analyzes are non-destructive and do not require sample preparation. Also, specific temperature conditions is not a requirement for NIR analysis. Please change this sentence.

Response 1: We sincerely thank the reviewer for pointing out our mistake. We have changed our text into "Firstly, in order to obtain NIRS data, the samples will be collected, stored and specially processed if necessary. Then each sample is measured with a near infrared spectrum instrument.  The obtained spectrum range is usually between 4000 and 10000 cm-1." in line 70-75.

Point 2: Lines 97-98 – “multiple scattering correction (MSC)” – MSC is multiplicative scatter correction.

Response 2: We sincerely thank the reviewer for pointing out this mistake. As suggested by the reviewer, we have corrected the “multiple scattering correction (MSC)” into “multiplicative scatter correction (MSC)” in line 104.

Point 3: Equations 1 and 2 are duplicated

Response 3: We were really sorry for our careless mistakes. The correction has been made in line 177-178 and the duplicated equations have been deleted.

Reviewer 3 Report

Clustering and classification are the major subdivisions of pattern recognition. Using these techniques, samples can be classified according to a specific property through measurements indirectly related to a property of interest. The manuscript submitted is a review on different discriminant methods, such as previously established ones, but also shedding light onto newly developed algorithms based on deep learning methods, all with special focus on the application of near-infrared spectroscopy for food quality assessment. There is an enormous amount of information within this review and it should be accepted after a minor revision.

A few items need alteration, such as on page 3:  “… has thousands of dimensions.  “, which must be thought over as the spectral resolution is often around 30 cm-1 and worse, in particular true for the short-wave NIR spectral range with even much broader absorption bands. In addition, preprocessing such as variable selection can reduce the dimensionality significantly. Also what are “complex noises” in the context of smoothing?

My suggestions also include implementing some recent literature, for example, dealing with pattern recognition as mentioned above. What about use of random forest (RF) classifiers, which I missed in this review?

Richard G. Brereton, Pattern recognition in chemometrics, Chemometrics and Intelligent Laboratory Systems 149 (2015) 90–96

As preprocessing is an important tool, the following reviews should be considered (not only important for regression but also for discriminant analysis methods; unfortunately, Ref. 9 was not traceable by literature search):

  1. Rinnan, F. van den Berg, S. Balling Engelsen, Review of the most common pre-processing techniques for near-infrared spectra, Trends in Analytical Chemistry 28 (10), 1201 (2009)
  2. Oliveri, C. Malegori, R. Simonetti, M. Casale, The impact of signal pre-processing on the final interpretation of analytical outcomes - A tutorial, Analytica Chimica Acta 1058, 9-17 (2019)
  3. Jiao, Z. Li, X. Chen, S. Fei, Preprocessing methods for near‐infrared spectrum calibration, Journal of Chemometrics 34 (11) e3306., first published: 15 October 2020 https://doi.org/10.1002/cem.3306

Yong-Huan Yun, Hong-Dong Li, Bai-Chuan Deng, Dong-Sheng Cao, An overview of variable selection methods in multivariate analysis of near-infrared spectra, Trends in Analytical Chemistry 113 (2019) 102-115

Further comments:

Lettering size in Figure 1 is partly too small to be readable

p.1, line 21: replace “its” by “their”

p.9: SIMCA is misspelled

p.6, , line 198: please explain VC dimension

p. 12, line 447: “clearly” is misspelled

Author Response

Response to Reviewer 3 Comments

Point 1: A few items need alteration, such as on page 3: “… has thousands of dimensions.  “, which must be thought over as the spectral resolution is often around 30 cm-1 and worse, in particular true for the short-wave NIR spectral range with even much broader absorption bands. In addition, preprocessing such as variable selection can reduce the dimensionality significantly. Also what are “complex noises” in the context of smoothing?

Response 1: We sincerely appreciate the reviewer for clarifying this for us. We have deleted the wrong expression and the sentence read as: "The obtained spectrum range is usually between 4000 and 10000 cm-1" in line 74-75. As for the “complex noises” part, we apologize for our poor writing, and we have changed this expression into "Although the near-infrared spectrometer is equipped with high-performance detection device, the collected data can still be affected by the noise of the measuring instrument, thus affecting the effect of modeling" in line 75-78.

Point 2: My suggestions also include implementing some recent literature, for example, dealing with pattern recognition as mentioned above. What about use of random forest (RF) classifiers, which I missed in this review?

Richard G. Brereton, Pattern recognition in chemometrics, Chemometrics and Intelligent Laboratory Systems 149 (2015) 90–96

Response 2: We sincerely appreciate the valuable comments. We have checked this literature carefully and added in the revised manuscript in line 134.

[23] Brereton, R. G. Pattern recognition in chemometrics. Chemometer. Intell. Lab. 2015, 149, 90-96.

In addition, the application of random forest (RF) classifiers was also included in the revised manuscript in line 378-387.

"For example, the random forest (RF) and PCA were used to establish the identification model of green tea species based on the joint information from NIRS and ultravio-let-visible spectroscopy [86]. The results show that the RF model can effectively iden-tify five green tea varieties, and the accuracy rate of the model was 96%. Similarly, in [87], Fourier transform mid-infrared spectroscopy and NIRS technologies were com-bined with single spectra analysis and multi-sensor information fusion strategy to es-tablish the RF model for the origin identification of Panax notoginseng. The accuracy of RF identification model was 95.6%, suggesting an effective identification ability to identify Panax notoginseng from five different origins."

Point 3: As preprocessing is an important tool, the following reviews should be considered (not only important for regression but also for discriminant analysis methods; unfortunately, Ref. 9 was not traceable by literature search):

Rinnan, F. van den Berg, S. Balling Engelsen, Review of the most common pre-processing techniques for near-infrared spectra, Trends in Analytical Chemistry 28 (10), 1201 (2009)

Oliveri, C. Malegori, R. Simonetti, M. Casale, The impact of signal pre-processing on the final interpretation of analytical outcomes - A tutorial, Analytica Chimica Acta 1058, 9-17 (2019)

Jiao, Z. Li, X. Chen, S. Fei, Preprocessing methods for near‐infrared spectrum calibration, Journal of Chemometrics 34 (11) e3306., first published: 15 October 2020 https://doi.org/10.1002/cem.3306

Yong-Huan Yun, Hong-Dong Li, Bai-Chuan Deng, Dong-Sheng Cao, An overview of variable selection methods in multivariate analysis of near-infrared spectra, Trends in Analytical Chemistry 113 (2019) 102-115

Response 3: We sincerely appreciate the reviewer for pointing this out. As suggested by the reviewer, we have added references to support the content, and Ref. 9 has been replaced.

[10] Rinnan, Å.; Van Den Berg, F.; Engelsen, S. B. Review of the most common pre-processing techniques for near-infrared spectra. Trends Analt. Chem. 2009, 28(10), 1201-1222.

[11] Jiao, Y.; Li, Z.; Chen, X.; Fei, S. Preprocessing methods for near‐infrared spectrum calibration. J. Chemometr., 2020, 34(11), e3306.

[12] Yun, Y. H.; Li, H. D.; Deng, B. C.; Cao, D. S. An overview of variable selection methods in multivariate analysis of near-infrared spectra. Trends Analt. Chem. 2019, 113, 102-115.

Point 4: Further comments:

Lettering size in Figure 1 is partly too small to be readable

p.1, line 21: replace “its” by “their”

p.9: SIMCA is misspelled

p.6, line 198: please explain VC dimension

  1. 12, line 447: "clearly" is misspelled

Response 4: We sincerely thank the reviewer for careful reading and sorry for our carelessness.

The letter in Figure 1 has been changed into readable size.

The word "its" has been replaced by "their" in p.1, line 22.

The misspelled "SIMCA" has been corrected in p. 6, line 185

The explanation has been made in p. 6, line 206-207, the text reads:" The VC dimension for the set of functions {F} is defined as the maximum number of training points which can be shattered by {F} [46]."

The misspelled "cleary" has been corrected into "clearly".

Round 2

Reviewer 1 Report

All the comments have been taken into account, therefore I recommend accepting the manuscript.

This manuscript is a resubmission of an earlier submission. The following is a list of the peer review reports and author responses from that submission.

Round 1

Reviewer 1 Report

This article covers an interesting topic but cannot be properly reviewed and should not be published until the English language and style are corrected.

In addition, in the first few sections, authors make a number of claim that are just not true. I made a detailed review until line 100 but then stopped as there were too many style and English issues.

Lines 38-39 – “Near-infrared spectroscopy (NIRS) technology is a newly developed detection tool” – NIRS is not new. It has been around since the early 1980s.

Line 42 – “recognition model”. Authors should generalize and use empirical instead of recognition

Figure 1 – “Data divided” – maybe replace by data handling?

Lines 57-64 – “Firstly, to obtain the NIR spectral data, all solid samples are crushed, weighed, and packaged in quartz cuvettes separately. All liquid samples are directly measured in standard quartz cuvettes. Then, under specific temperature and humidity conditions, each powder sample is measured using a near-infrared spectrometer equipped with an integrating sphere and a rotating sample cup. Obtained spectral range is usually from 4000 to 10000 cm-1 and is of thousands of dimensions. Firstly, although the spectrometer is equipped with a high-sensitivity InGaAs detector and automatic gold foil background collector, complex noise contained in spectrum cannot be eliminated.” – these statements are not a correct representation of the hundreds of applications of NIR in food. Authors must make these statements more general. Not all solid samples are crushed, weighed and packed. Not all spectrometers use InGaAs detectors… please stay more general.

Lines 73-75 – “It should be mentioned that the pattern recognition method is an important branch of chemometrics, which is more popular and has higher precision than traditional chemometrics” – these claims are not credible. Please refrain from making such general and unproven statements.

Figure 2 – sample is misspelled. Replace “divided dada” by a more suitable term.

Line 92 – “the symbol of scatter correction” – symbol is not an appropriate word here

Lines 94-95 – “Scale scaling can eliminate the adverse effects caused by large scale differences, including mean centralization” – scale scaling is not proper language. Do authors wanted to state mean centering instead of centralization?

Reviewer 2 Report

The authors provide a nice and interesting review. In this paper, the currently developed NIRS based food discrimination methods are reviewed.

The review discusses most of the key publications relevant to the topic. However, I suggest to include a section about discrimination of food using NIR hyperspectral imaging and multivariate data analysis in the review.

Also, I have only a few comments or suggestions to help improve the quality of it.

There are a few grammar and misspelling errors that can be easily corrected:

Figure 2. Replace “Sampls” by “samples”

All the scientific names should be written in cursive in the manuscript